# Deficiency of orexin receptor type 1 in dopaminergic neurons increases novelty-induced locomotion and exploration

Xing Xiao[1]*, Gagik Yeghiazaryan[2,3], Fynn Eggersmann[2,3], Anna Lena Cremer[4], Heiko Backes[4], Peter Kloppenburg[2,3], Anne Christine Hausen[1]*

[1]Max Planck Institute for Metabolism Research, Department of Neuronal Control of Metabolism, Cologne, Germany; [2]Department of Biology, Institute for Zoology, University of Cologne, Cologne, Germany; [3]Excellence Cluster on Cellular Stress Responses in Aging Associated Diseases (CECAD) and Center for Molecular Medicine Cologne (CMMC), University of Cologne, Cologne, Germany; [4]Max Planck Institute for Metabolism Research, Multimodal Imaging of Brain Metabolism Group, Cologne, Germany

**\*For correspondence:**
xing.xiao@charite.de (XX);
christine.hausen@sf.mpg.de
(ACH)

**Competing interest:** The authors declare that no competing interests exist.

## eLife Assessment

This manuscript describes **valuable** findings regarding the expression pattern of orexin receptors in the midbrain and how manipulating this system influences several behaviors, such as context-induced locomotor activity and exploration. The overall strength of evidence - which includes anatomical, viral manipulation studies, and brain imaging - is **solid** and broadly substantiates claims in the paper. However, there are several areas in which the conclusions are only partially supported by the combination of methods used. These results have implications for understanding the neural underpinnings of reward and will be of interest to neuroscientists and cognitive scientists with an interest in the neurobiology of reward.

**Abstract** Orexin signaling in the ventral tegmental area and substantia nigra promotes locomotion and reward processing, but it is not clear whether dopaminergic neurons directly mediate these effects. We show that dopaminergic neurons in these areas mainly express orexin receptor subtype 1 (Ox1R). In contrast, only a minor population in the medial ventral tegmental area express orexin receptor subtype 2 (Ox2R). To analyze the functional role of Ox1R signaling in dopaminergic neurons, we deleted Ox1R specifically in dopamine transporter-expressing neurons of mice and investigated the functional consequences. Deletion of Ox1R increased locomotor activity and exploration during exposure to novel environments or when intracerebroventricularely injected with orexin A. Spontaneous activity in home cages, anxiety, reward processing, and energy metabolism did not change. Positron emission tomography imaging revealed that Ox1R signaling in dopaminergic neurons affected distinct neural circuits depending on the stimulation mode. In line with an increase of neural activity in the lateral paragigantocellular nucleus (LPGi) of Ox1R$^{\Delta DAT}$ mice, we found that dopaminergic projections innervate the LPGi in regions where the inhibitory dopamine receptor subtype D2 but not the excitatory D1 subtype resides. These data suggest a crucial regulatory role of Ox1R signaling in dopaminergic neurons in novelty-induced locomotion and exploration.

## Introduction

Orexin neurons are exclusively located in the lateral hypothalamus (LH) and adjacent regions in the brain, including the perifornical area, and dorsomedial and posterior hypothalamus (*Soya and Sakurai, 2020*). Abundant orexin fibers innervate the ventral tegmental area (VTA) and substantia nigra (SN), which together contain the majority of dopaminergic neurons in humans and rodents (*Fadel and Deutch, 2002*; *Hrabovszky et al., 2013*; *Peyron et al., 1998*). There are also some dopaminergic neurons in other brain regions, although not all of these neurons express the dopamine transporter (DAT) (*Koblinger et al., 2018*; *Sharples et al., 2014*; *Turiault et al., 2007*). Both the orexin and dopaminergic systems control food intake, locomotor activity, reward processing as well as energy metabolism (*Howe and Dombeck, 2016*; *Inutsuka and Yamanaka, 2013*; *Narayanan et al., 2010*; *Palmiter, 2007*; *Tsujino and Sakurai, 2013*). Dopamine was discovered as a neuro-active molecule approximately 70 y ago and the dopaminergic system has diverse functions (*Iversen and Iversen, 2007*). In a simplified model the nigrostriatal dopaminergic system, containing the dopaminergic projections from SN to the dorsal striatum, predominantly regulates action selection and exploratory behaviors. In contrast, the mesolimbic and mesocortical dopaminergic systems, which consist of the dopaminergic pathways from the VTA to the ventral striatum and prefrontal cortex, predominantly regulate motivation, cognition, decision-making, reward, and aversive behavior (*Kutlu et al., 2021*; *Latif et al., 2021*; *Lerner et al., 2021*; *Roeper, 2013*). Some VTA dopaminergic neurons also represent movements in trained animals (*Lerner et al., 2021*). Dopaminergic neurons modulate movement differently via direct and indirect pathways through the striatum (*Kim et al., 2017*; *Kreitzer and Malenka, 2008*; *Roseberry et al., 2016*). Co-release of other neuro-active molecules by dopaminergic terminals (*Barcomb and Ford, 2023*; *Buck et al., 2021*; *Zych and Ford, 2022*) and the spatial and temporal dynamics of dopamine release (*Berke, 2018*; *Chantranupong et al., 2023*; *Liu et al., 2021*) could further affect the precision and diversity of dopaminergic functions.

Orexin A injection into the VTA increases cocaine- and morphine-related reward processing (*Mahler et al., 2012*), and intra-SN injection of orexin A induces hyperlocomotion and stereotypic behaviors (*Liu et al., 2018*; *Nakamura et al., 2000*). On the other hand, systemic injection of dopamine receptor antagonists reduces orexin A-induced increase of hyperlocomotion, and stereotypic and grooming behaviors (*Nakamura et al., 2000*). However, whether the above effects of orexin signaling in the VTA and SN are mediated by dopaminergic or non-dopaminergic cells is not yet clear.

The orexin system consists of the two peptides orexin A and B, which are derived from the precursor, prepro-orexin (*de Lecea et al., 1998*; *Sakurai et al., 1998*). Both are endogenous ligands for two G-protein-coupled receptors, orexin receptor subtype 1 (Ox1R), and 2 (Ox2R) (*Sakurai et al., 1998*). Orexin A activates both receptors, while orexin B predominantly activates Ox2R (*Sakurai et al., 1998*). Detection of mRNA by in situ hybridization revealed the expression of both receptors in the VTA and SN (*Marcus et al., 2001*). However, the cell-type specific expression of Ox1R and Ox2R in the VTA and SN is not well understood. Electrophysiological studies showed that orexin receptors are present in dopaminergic VTA neurons (*Baimel et al., 2017*; *Korotkova et al., 2003*; *Tung et al., 2016*), whereas data for the SN seem controversial (*Korotkova et al., 2002*; *Liu et al., 2018*).

Here, we aim to map the expression patterns of Ox1R and Ox2R in dopaminergic VTA and SN neurons and investigate the functional role of Ox1R using conditional knockout mice.

## Results

### Ox1R is the predominant orexin receptor in dopaminergic neurons

To analyze the expression patterns of orexin receptors in dopaminergic VTA and SN neurons, we performed fluorescent in situ hybridization (RNAscope) of tyrosine hydroxylase (*Th*), Ox1R (*Hcrtr1*) and Ox2R (*Hcrtr2*) in control (Ox1R$^{fl/fl}$, DAT$^{wt/wt}$) mice and Ox1R$^{\Delta DAT}$ (Ox1R$^{fl/fl}$, DAT-Cre$^{tg/wt}$) mice, from the same breeding crossing mice carrying a loxP-flanked Ox1R-allele (Ox1R$^{fl/fl}$) with DAT-Cre$^{tg/wt}$ mice (*Ekstrand et al., 2007*).

In control mice, Ox1R was expressed in 44–58% of dopaminergic neurons in all VTA and SN subregions, including the interfascicular (IF), paranigral (PN), parainterfascicular (PIF), and parabrachial (PBP) subnuclei of the VTA and SN. In contrast, Ox2R-positive dopaminergic neurons were mainly located in the medial VTA, including IF, PN, and PIF, with 25–42% of dopaminergic neurons expressing the Ox2R (*Figure 1A, C, D, G-I*; *Figure 1—figure supplement 1A*). While the above data were

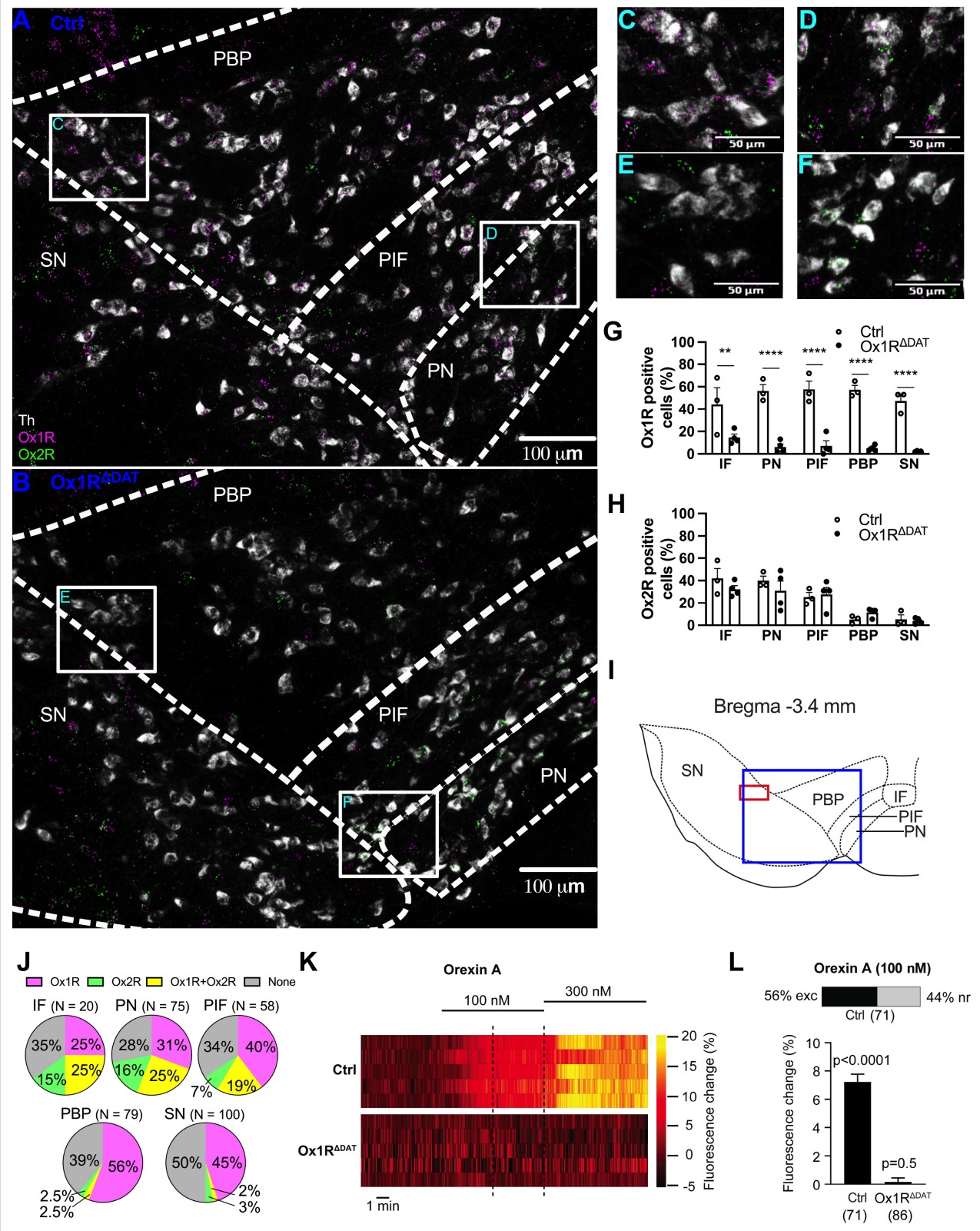

**Figure 1.** Orexin receptor subtype 1 (Ox1R) predominantly mediates orexin-induced activation of dopaminergic neurons in the substantia nigra (SN). Representative images of RNAscope in situ hybridization in SN and paranigral (PN), parainterfascicular (PIF), and parabrachial (PBP) subnuclei of the ventral tegmental area (VTA), of (**A**) a control and (**B**) an Ox1R$^{\Delta DAT}$ mouse. Amplifications of the (**C**) left and (**D**) right marked areas in white squares in (**A**). Amplifications of the (**E**) left and (**F**) right marked area in white squares in (**B**). White, tyrosine hydroxylase (Th); magenta, Ox1R; green, Ox2R. Scale

*Figure 1 continued on next page*

*Figure 1 continued*

bar: 100 μm (**A, B**) or 50 μm (**C–F**). Representative images of an overview in SN and VTA were shown in *Figure 1—figure supplement 1*. Percentages of (**G**) Ox1R and (**H**) Ox2R positive neurons in Th positive neurons. IF, interfascicular subnuclei of VTA. Control, n=3; Ox1R$^{\Delta DAT}$, n=4. Data are represented as means ± SEM. **p<0.01; ****p<0.0001; as determined by two-way ANOVA followed by Sidak's post hoc test. (**G**) Subnuclei: $F_{(4, 25)}$=0.60, p=0.66; genotype: $F_{(1, 25)}$=173.4, p<0.0001; subnuclei x genotype: $F_{(4, 25)}$=1.43, p=0.25. (**H**) Subnuclei: $F_{(4, 25)}$=17.06, p<0.0001; genotype: $F_{(1, 25)}$=0.50, p=0.48; subnuclei x genotype: $F_{(4, 25)}$=0.90, p=0.48. (**I**) Schematic illustration showing the subnuclei of the left VTA and SN. The blue frame indicates the region where panels (**A**) and (**B**) are showing and the red frame indicates the region where calcium imaging was performed. (**J**) Pie charts showing Ox1R and Ox2R expression in dopaminergic neurons in the VTA and SN. Three control mice were analyzed and N indicates the mean dopaminergic neuron numbers in the respective subregions. Magenta, dopaminergic neurons only expressing Ox1R; green, dopaminergic neurons only expressing Ox2R; yellow, dopaminergic neurons expressing both Ox1R and Ox2R; gray, dopaminergic neurons expressing neither Ox1R or Ox2R. (**K, L**) Effect of orexin A on dopaminergic SN neurons analyzed by Ca$^{2+}$ imaging with GCaMP6s. Recordings were performed in acute brain slices from control and Ox1R$^{\Delta DAT}$ male mice with GCaMP6s expressed in dopaminergic SN neurons. (**K**) Exemplary heat maps of five individual orexin A responsive dopaminergic SN neurons from control (top) and five not orexin A-responsive dopaminergic SN neurons of Ox1R$^{\Delta DAT}$ mice (bottom). The recordings show the responses to 100 nM and 300 nM orexin A. The dashed lines indicate the range where the responses to 100 nM were quantified. Heat maps of all recorded neurons are shown in *Figure 1—figure supplement 4*. (**L**) Top: The stacked bar shows the percentage of individual dopaminergic neurons in control mice in which the increase in [Ca$^{2+}$]$_i$ was larger than three times the standard deviation of the baseline fluorescence (3 σ criterion ≙ Z-score of 3), thus defining them as orexin A responsive (see Materials and methods). The dopaminergic neurons in Ox1R$^{\Delta DAT}$ mice did not respond to 100 nM orexin A. Bottom: Population Ca$^{2+}$ responses upon 100 nM orexin A application from all recorded dopaminergic SN neurons of control and Ox1R$^{\Delta DAT}$ mice. Data are shown as the percentage of the maximal response to high K$^+$ saline. The significance of this mean response was tested for each group (control and Ox1R$^{\Delta DAT}$) using one-sample t-tests (control: p<0.0001, n=71; Ox1R$^{\Delta DAT}$: p=0.5, n=86). Bar graphs represent means ± SEM. p-values are provided above the bar graphs. n-values are given in brackets below the bar graphs. The full statistics are provided alongside the source data.

The online version of this article includes the following figure supplement(s) for figure 1:

**Figure supplement 1.** An overview of orexin receptor expression in the substantia nigra (SN) and the ventral tegmental area (VTA).

**Figure supplement 2.** Orexin A increases the action potential frequency of dopaminergic neurons in the substantia nigra (SN).

**Figure supplement 3.** Orexin A and high K$^+$ saline effect on [Ca$^{2+}$]$_i$ of dopaminergic neurons in the substantia nigra (SN) of control and Ox1R$^{\Delta DAT}$ mice which express GCaMP6s.

**Figure supplement 4.** Orexin A and high K$^+$ saline effect on [Ca$^{2+}$]$_i$ of dopaminergic neurons in the substantia nigra (SN) of control and Ox1R$^{\Delta DAT}$ mice which express GCaMP6s.

calculated as the mean percentages in the investigated mice, we also calculated the mean dopaminergic neuron numbers separately for those expressing both or exclusively one of the orexin receptor types. Thus, we could analyze the distribution of Ox1R-, Ox2R-, Ox1R, and Ox2R-, and none (of orexin receptors)-expressing dopaminergic neurons in each VTA subregion and the SN (*Figure 1J*). The results were similar to the above: 47–59% of dopaminergic neurons expressed Ox1R in all VTA and SN sub-regions, and Ox2R was mainly expressed by 26–41% of dopaminergic neurons in the medial VTA (*Figure 1J*). 19–25% of dopaminergic neurons in the medial VTA expressed both Ox1R and Ox2R (*Figure 1J*). Of note, abundant Ox1R and Ox2R expression was present in Th-negative neurons in the VTA, SN, and neighboring regions (*Figure 1A, C, D*, *Figure 1—figure supplement 1A*).

Since most dopaminergic neurons express Ox1R, we focused the functional analysis on Ox1R signaling in dopaminergic neurons. Therefore, Ox1R was specifically inactivated in dopaminergic (DAT-expressing) neurons of mice (Ox1R$^{\Delta DAT}$). As a prerequisite for the functional analysis (behavior and PET imaging) validation of the Ox1R deletion was necessary. In Ox1R$^{\Delta DAT}$ mice, Ox1R expression was significantly reduced in dopaminergic neurons and remained unchanged in Th-negative cells (*Figure 1B, E-I*; *Figure 1—figure supplement 1B*). In contrast, the Ox2R expression patterns remained similar to that in the control mice, indicating the successful and specific knockout of Ox1R in dopaminergic neurons (*Figure 1B, E, I*; *Figure 1—figure supplement 1B*).

Next, we used perforated patch clamp recordings to establish that orexin A activates dopaminergic neurons not only in the mouse VTA (e.g. *Baimel et al., 2017*; *Korotkova et al., 2003*; *Tung et al., 2016*), but also in the mouse SN (*Figure 1—figure supplement 2*). In dopaminergic SN neurons that were pharmacologically isolated from GABAerigc and glutamatergic synaptic input 100 nM orexin increased the action potential frequency in 75% (6 of 8) of the neurons. Increasing the orexin A concentration to 300 nM further increased the firing frequency. This finding is consistent with previous extracellular recordings in rats (*Liu et al., 2018*). To also validate the Ox1R knockout in dopaminergic neurons functionally, we performed Ca$^{2+}$ imaging in acute brain slices from control and Ox1R$^{\Delta DAT}$ mice. We used the genetically encoded Ca$^{2+}$ indicator GCaMP6s to monitor [Ca$^{2+}$]$_i$ specifically in dopaminergic (DAT-expressing) neurons. This analysis focused on

dopaminergic neurons in the SN, which we used as an 'indicator population' because a large number of these neurons express Ox1R. During the experiments, GABAergic and glutamatergic synaptic input was blocked and orexin A was bath-applied at 100 nM and 300 nM. Bath application of 100 nM orexin A already resulted in a clear and significant increase in $[Ca^{2+}]_i$ in dopaminergic SN control neurons (*Figure 1I-L*, *Figure 1—figure supplements 3 and 4*). For quantification, we tested each neuron whether it responded to 100 nM orexin A. A neuron was considered orexin-responsive if the change in GCaMPs fluorescence induced by orexin A was three times larger than the baseline fluorescence's standard deviation (3 σ criterion ≙ Z-score of 3). We found that 56% of the neurons tested responded to 100 nM orexin A, while 44% of the neurons did not respond (*Figure 1L*, *Figure 1—figure supplement 4*). These data are in very good agreement with the number of Ox1R-expressing neurons (*Figure 1J*). To determine the 'population response' of all analyzed neurons, we measured the orexin-induced GCaMP6s fluorescence of each neuron expressed as a percentage of the GCaMP6s fluorescence induced by applying high (40 mM) $K^+$ saline (*Figure 1L*, *Figure 1—figure supplement 3*). This is a solid reference point since it reflects the GCaMP6s fluorescence at maximal voltage-activated $Ca^{2+}$ influx. The 'population response' of all analyzed neurons was expressed as the mean ± SEM of these responses (*Figure 1L*). The significance of this mean response was tested for each group (control and Ox1R$^{\Delta DAT}$) using a one-sample t-test. We found a highly significant and marked response of control neurons to 100 nM orexin A, while the Ox1R knockout neurons did not respond (control: p<0.0001, n=71; Ox1R$^{\Delta DAT}$: p=0.5, n=86; one sample t-test for each group) (*Figure 1L*).

In conclusion, Ox1R is the main orexin receptor expressed in dopaminergic VTA and SN neurons and mediates their activation by orexin. Therefore, we wanted to determine further the functional role of Ox1R signaling in dopaminergic neurons. By comparing control and Ox1R$^{\Delta DAT}$ mouse data, we first identified behaviors modulated by orexin via dopaminergic neurons. In a second step, we used PET imaging to define and map neuronal pathways modulated by orexin.

## Ox1R in dopaminergic neurons regulates novelty-induced locomotion and exploration

The behavioral test included an analysis of locomotor behavior, novelty exploration, reward processing, anxiety, and energy homeostasis. Since sex differences exist in orexin- and dopamine-mediated behaviors (*Durairaja and Fendt, 2021*; *Freeman et al., 2021*; *Lippert et al., 2020*; *Zachry et al., 2021*), we analyzed behavioral phenotypes in both males and females. Compared to controls, Ox1R$^{\Delta DAT}$ mice did not show significant changes in spontaneous locomotor activity in home cages (*Figure 2—figure supplement 1*). When exposed to a novel open field, male and female Ox1R$^{\Delta DAT}$ mice exhibited an increase in locomotion and exploration, with increased traveling distance, ambulating time, and vertical activity in the open field test (*Figure 2A–C and E–G*). Female Ox1R$^{\Delta DAT}$ mice also showed increased stereotypic activity (*Figure 2D and H*). Furthermore, intracerebroventricular (ICV) injection of orexin A (1 nmol) induced a more pronounced increase of locomotor activity in both female and male Ox1R$^{\Delta DAT}$ mice, compared to control mice (*Figure 2I–L*).

In a hole-board test, female Ox1R$^{\Delta DAT}$ mice showed increased nose pokes into the holes in early (1st and 2nd) sessions compared to control mice (*Figure 2—figure supplement 2B*). This effect disappeared later when mice became familiar with the environment (third – eighth session) (*Figure 2—figure supplement 2B*). In the novel object recognition test, female Ox1R$^{\Delta DAT}$ mice spent more time with each of the two identical objects in an open field compared to control mice (*Figure 2—figure supplement 2E*). One hour later, when female mice were exposed to the same environment except that one object was replaced by a novel object, there was no significant difference between control and Ox1R$^{\Delta DAT}$ mice in exploring the novel object (*Figure 2—figure supplement 2F*). In contrast, male Ox1R$^{\Delta DAT}$ mice did not behave differently in the hole-board and novel object recognition tests from controls (*Figure 2—figure supplement 2A, C, D*). This indicates that Ox1R signaling in dopaminergic neurons is more important for females in novelty-induced exploration, which was independent of novel object discrimination.

The behavioral changes seemed not related to alterations in anxiety since there were no detectable changes in anxiety-related behaviors, including the time spent in and the times of entries into the center area, light side, and the open arms in an open field test, dark/light box test and 0-maze test, respectively (*Figure 2—figure supplement 3*).

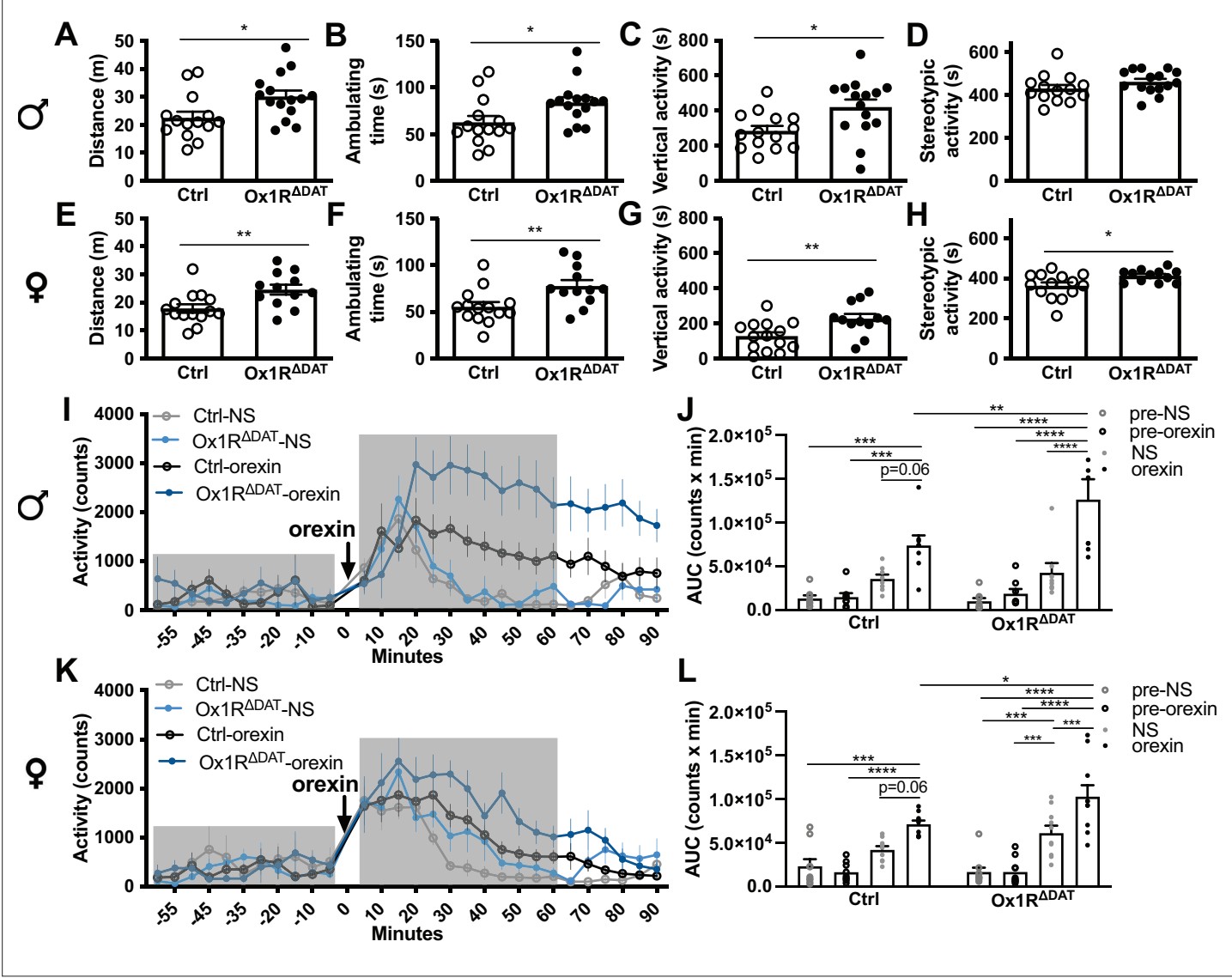

**Figure 2.** Deficiency of orexin receptor subtype 1 (Ox1R) in dopaminergic neurons increases locomotion and exploration behaviors. (**A**) Total traveling distance, (**B**) ambulating time, (**C**) vertical activity, and (**D**) stereotypic activity of male mice in the open field test. Control, n=14; Ox1R$^{\Delta DAT}$, n=15. (**E**) Total traveling distance, (**F**) ambulating time, (**G**) vertical activity, and (**H**) stereotypic activity of female mice in the open field test. Control, n=14; Ox1R$^{\Delta DAT}$, n=12. Locomotor activity upon intracerebroventricular (ICV) injection of saline (NS) or orexin A in (**I**) male and (**K**) female control and Ox1R$^{\Delta DAT}$ mice. Gray boxes indicate the pre- and post- injection period for the area under curve (AUC) quantification. (**J, L**) Quantification of AUC pre- and post-saline and orexin A injection to (**J**) male and (**L**) female mice. Male (control and Ox1R$^{\Delta DAT}$), n=8; female-control, n=9; female-Ox1R$^{\Delta DAT}$, n=10. Data are represented as means ± SEM. *p<0.05; **p<0.01; ***p<0.001; ****p<0.0001; as determined by unpaired two-tailed Student's t-test (**A–H**), or mixed two-way ANOVA followed by Sidak's post hoc test (**J, L**). (**J**) Treatment: $F_{(1, 14)}=2.34$, p=0.15; genotype: $F_{(3, 42)}=38.59$, p<0.0001; treatment × genotype: $F_{(3, 42)}=3.76$, p=0.018; subject: $F_{(14, 42)}=2.27$, p=0.02. (**L**) Treatment: $F_{(1, 17)}=3.73$, p=0.084; genotype: $F_{(3, 51)}=45.52$, p<0.0001; treatment × genotype: $F_{(3, 51)}=3.23$, p=0.030; subject: $F_{(17, 51)}=1.55$, p=0.12. The full statistics are provided alongside the source data.

The online version of this article includes the following figure supplement(s) for figure 2:

**Figure supplement 1.** Unaltered mouse spontaneous activity in home cages by deficiency of orexin receptor subtype 1 (Ox1R) in dopaminergic neurons.

**Figure supplement 2.** Increased exploratory behaviors of female Ox1R$^{\Delta DAT}$ in the hole-board (HB) test and the novel object test (NOT).

**Figure supplement 3.** Unaltered anxiety-related behaviors in Ox1R$^{\Delta DAT}$ mice.

**Figure supplement 4.** Unaltered reward-related behaviors in Ox1R$^{\Delta DAT}$ mice.

**Figure supplement 5.** Unaltered energy homeostasis in Ox1R$^{\Delta DAT}$ mice.

## Ox1R deletion in dopaminergic neurons did not change energy metabolism and reward processing

The orexin and dopaminergic systems are well known to regulate energy metabolism and reward processing (*Howe and Dombeck, 2016*; *Inutsuka and Yamanaka, 2013*; *Narayanan et al., 2010*; *Palmiter, 2007*; *Tsujino and Sakurai, 2013*). Therefore, we measured the impact of Ox1R deletion in dopaminergic neurons on metabolism and reward-related behaviors.

In the two-bottle preference test, mice have free access to both water and sweet solution (sucrose solution or the non-caloric sweetener, sucralose solution). In both of the control and Ox1R$^{\Delta DAT}$ groups, the percentages of the consumption of 1% sucrose, 2% sucrose, and 0.04% sucralose solution in the total drinking were higher than 50%, indicating the preference for sweet solution, though only female control mice showed a significant preference of 0.5% sucrose solution, analyzed by a one-sample t-test (*Figure 2—figure supplement 4A, D*). There was no significant difference between the two groups by two-way ANOVA followed by Sidak's post hoc test (*Figure 2—figure supplement 4A, B, D, E*).

For the conditional place preference test (CPP, cocaine- vs. saline-injection-paired compartment), the CPP score was calculated to show how much cocaine injection changed the time that mice spent in the compartment. All groups except the male control group showed a significant preference for cocaine injection using a one-sample t-test, and there was no significant difference between control and Ox1R$^{\Delta DAT}$ groups by unpaired two-tailed Student's t-test (*Figure 2—figure supplement 4C, F*).

In addition, neither male nor female Ox1R$^{\Delta DAT}$ mice showed significant changes in energy metabolism, including body weight, fat mass, respiratory exchanging ratio, energy expenditure, daily food intake, glucose tolerance, and insulin sensitivity (*Figure 2—figure supplement 5*).

## Ox1R in dopaminergic neurons regulates specific neural pathways under different conditions

To explore and map the underlying neuronal pathways of the orexin-mediated behaviors unbiasedly, we performed positron emission tomography (PET) imaging of control and Ox1R$^{\Delta DAT}$ mice upon ICV injection of orexin A or saline (control).

In Ox1R$^{\Delta DAT}$ mice under control conditions (saline injection), we observed higher neuronal activity around the medial preoptic area (MPA), piriform cortex (Pir), endopiriform claustrum, lateral stripe of the striatum (LSS), ventral part of subcoeruleus nucleus (SubCV), spinal trigeminal nucleus, spinal trigeminal tract (sp5), the intermediate reticular zone (IRt) and the gigantocellular reticular nucleus (Gi) than control mice (*Figure 3A*, *Figure 3—figure supplements 1A and 2*). Compared to control mice, central application of orexin A (1 nmol, ICV) to Ox1R$^{\Delta DAT}$ mice induced higher neuronal activity around the LPGi, the nucleus of the horizontal limb of the diagonal band (HDB) and magnocellular preoptic nucleus (MCPO), and lower activity around secondary motor cortex (M2), the dorsal bed nucleus of the stria terminalis (BNST), the hindlimb, shoulder, and trunk regions and barrel field of primary somatosensory cortex (S1HL, S1Sh, S1Tr, and S1BF), the lateral area of secondary visual cortex (V2L), primary visual cortex (V1), Pir and dorsal endopiriform claustrum (DEn) (*Figure 3B*, *Figure 3—figure supplements 1B and 2*). PET imaging did not reveal significant changes in the VTA and SN (*Figure 3*, *Figure 3—figure supplement 1*).

Postmortem analysis of c-Fos staining revealed low c-Fos expression in dopaminergic neurons in the VTA and SN of Ox1R$^{\Delta DAT}$ and control mice after ICV injection of saline or orexin A (1 nmol). No apparent changes were observed among the groups. In contrast, clear orexin-induced c-Fos activity was observed in non-dopaminergic cells (*Figure 3—figure supplement 3*). This could affect the interpretation of dopaminergic contribution to behavioral effects in studies using ICV or intra-VTA/SN injection of orexin. Further studies would be needed to investigate how orexin directly and indirectly affects the VTA and SN neurons.

## Dopamine receptors in the dorsal BNST and LPGi

Based on the PET imaging and behavioral findings, we further focused on the dorsal BNST and LPGi. They play a key functional role in regulating emotion, exploratory behaviors, and locomotor speed, which are related to novelty-induced locomotion (*Arber and Costa, 2022*; *Avery et al., 2016*; *Capelli et al., 2017*; *Crestani et al., 2013*; *Giardino et al., 2018*; *Takatoh et al., 2013*). A schematic illustration of these analyzed brain areas is shown in *Figure 4—figure supplement 1*. We analyzed the

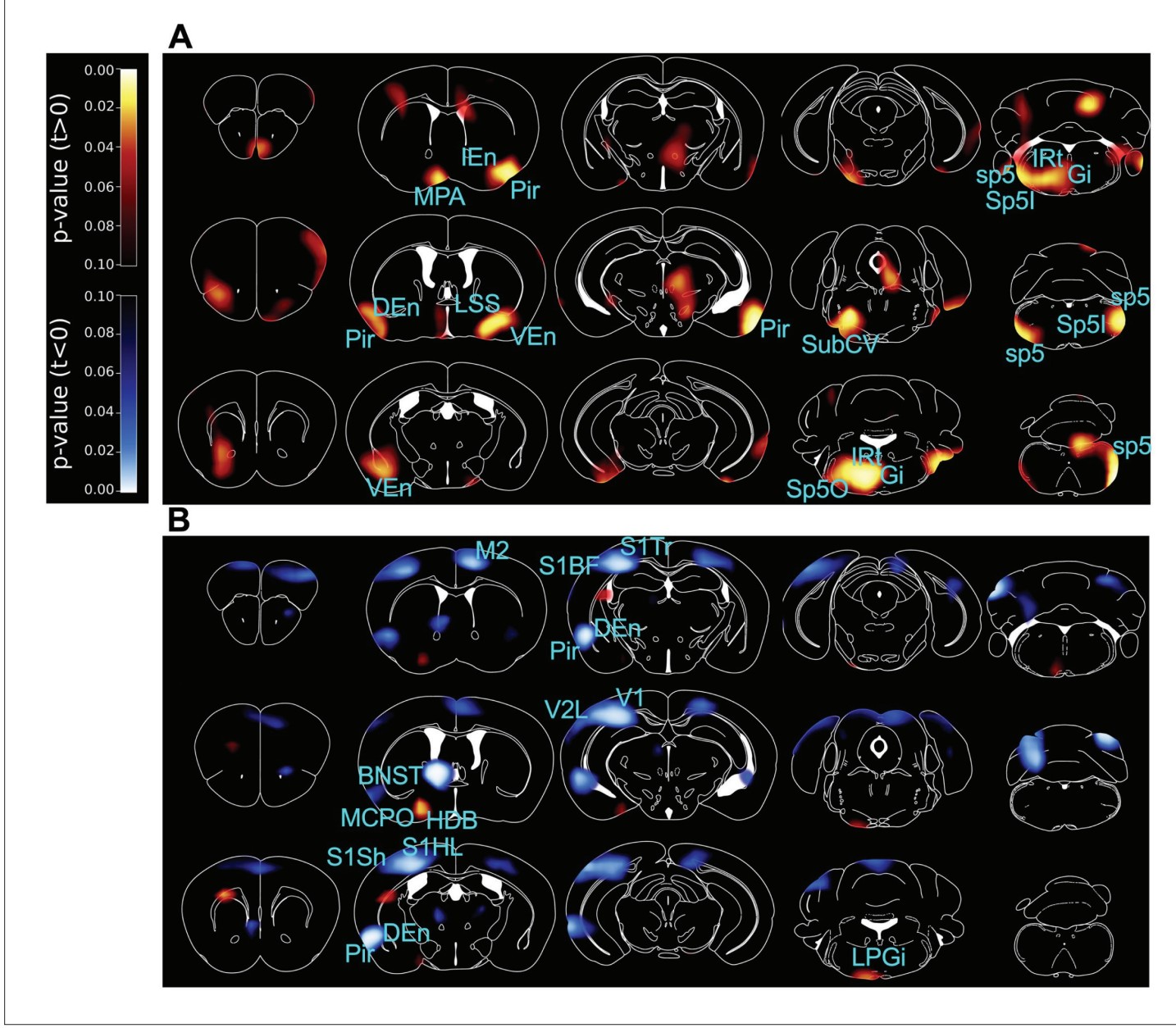

**Figure 3.** PET imaging studies comparing Ox1R$^{\Delta DAT}$ and control mice. 3D maps of p-values in positron emission tomography (PET) imaging studies comparing Ox1R$^{\Delta DAT}$ and control mice, after intracerebroventricular (ICV) injection of (**A**) saline (NS) and (**B**) orexin A. Brain areas with significant changes are indicated. Control-NS, n=8; control-orexin, n=6; Ox1R$^{\Delta DAT}$ (NS and orexin), n=8. Data are compared by unpaired two-tailed Student's t-test. M2, secondary motor cortex; MPA, medial preoptic area; Pir, piriform cortex; IEn, intermediate endopiriform claustrum; DEn, dorsal endopiriform claustrum; VEn, ventral endopiriform claustrum; LSS, lateral stripe of the striatum; BNST, the dorsal bed nucleus of the stria terminalis; HDB, nucleus of the horizontal limb of the diagonal band; MCPO, magnocellular preoptic nucleus; S1Sh, primary somatosensory cortex, shoulder region; S1HL, primary somatosensory cortex, hindlimb region; S1BF, primary somatosensory cortex, barrel field; S1Tr, primary somatosensory cortex, trunk region; V1, primary visual cortex; V2L, secondary visual cortex, lateral area; SubCV, subcoeruleus nucleus, ventral part; Gi, gigantocellular reticular nucleus; IRt, intermediate reticular nucleus; LPGi, lateral paragigantocellular nucleus; Sp5O, spinal trigeminal nucleus, oral part; Sp5I, spinal trigeminal nucleus, interpolar part; sp5, spinal trigeminal tract. More statistical information is provided in *Figure 3—figure supplements 1 and 2* and in the source data.

The online version of this article includes the following figure supplement(s) for figure 3:

**Figure supplement 1.** Positron emission tomography (PET) imaging studies comparing Ox1R$^{\Delta DAT}$ and control mice, shown as p-map images with the original voxel size.

**Figure supplement 2.** Statistical testing for the volumes of interest from brain regions in positron emission tomography (PET) imaging studies.

**Figure supplement 3.** c-Fos and tyrosine hydroxylase (Th) staining in substantia nigra (SN) and ventral tegmental area (VTA).

dopaminergic pathways into the dorsal BNST and LPGi that could have been affected by the Ox1R. To this end, we injected Cre-dependent AAV-EYFP bilaterally into the VTA and SN of DAT-Cre (DAT[EYFP]) and Ox1R[ΔDAT] (DAT[ΔOx1R; EYFP]) mice, and discovered dopaminergic projections in the dorsal BNST and LPGi (*Figure 4—figure supplement 2*), suggesting that VTA and SN dopaminergic neurons could directly modulate neuronal activity in the dorsal BNST and LPGi. To further explore this idea, we assessed the expression of Fos and dopamine receptors in the dorsal BNST and LPGi by immunostaining and RNAscope experiments.

Orexin A injection (1 nmol, ICV) induced higher c-Fos expression levels in LPGi of male Ox1R[ΔDAT] mice compared to control mice (*Figure 4A and C*). Only the inhibitory D2 but not excitatory D1 subtypes of dopamine receptors (DRD2, DRD1) could be detected in LPGi (*Figure 4D and E*). Both receptors were detected around the lateral ventricle (as positive control), whereas no signal was detectable in LPGi incubated with a negative control probe for mice (*Figure 4F and G*). Th-positive fibers were detected in LPGi but not Gi (*Figure 4A and B*), consistent with previous reports (*Kitahama et al., 2000*). These data provide the anatomical basis for the possibility that Ox1R deletion in dopaminergic neurons results in the disinhibition of neural activity in LPGi via dopaminergic pathways.

Orexin A injection increased c-Fos expression in the dorsal BNST in control but not in Ox1R[ΔDAT] mice (*Figure 4H and J*). The expression of DRD1 was significantly reduced, and DRD2 expression levels showed a tendency towards a decrease in dorsal BNST of Ox1R[ΔDAT] mice (*Figure 4K-M*, *Figure 4—figure supplement 3*). The Th expression levels were not changed between genotypes (*Figure 4H, I*, *Figure 4—figure supplement 3*). This indicates that the dorsal BNST could be less activated due to the decrease of DRD1 and the lack of orexin-mediated activation of dopaminergic neurons.

## Discussion

This study's behavioral experiments provide important evidence that Ox1R deficiency in DAT-expressing neurons leads to an increase in novelty-induced locomotion and exploration without altering energy metabolism and anxiety- or reward-processing-related behaviors. We found that Ox1R is the main subtype of orexin receptors in DAT-expressing neurons in VTA and SN, though the expression patterns of orexin receptors in DAT-expressing cells in brain areas outside VTA/SN remain unclear. The essential prerequisites for these functional studies were RNAscope and ex vivo Ca²⁺ imaging experiments, which confirmed the successful knockout of Ox1R in dopaminergic neurons. Further PET imaging, along with c-Fos and DRD expression analyses, provided initial insights into the underlying mechanisms. These findings can serve as a basis for developing testable working hypotheses on the systemic physiological role of orexin signaling.

Earlier studies showed orexin-induced activation of VTA dopaminergic neurons (*Baimel et al., 2017*; *Korotkova et al., 2003*; *Tung et al., 2016*). Whole-cell patch clamp recordings combined with mRNA measurement of the individually recorded VTA neurons revealed fewer Ox2R- than Ox1R-positive dopaminergic neurons (*Korotkova et al., 2003*), which is in line with our findings. While our electrophysiological and Ca²⁺ imaging data clearly show an orexin A-induced activation of dopaminergic SN neurons when GABAergic and glutamatergic synaptic input is blocked, previous in vitro (*Korotkova et al., 2002*) or in vivo extracellular electrophysiological recordings in rats (*Liu et al., 2018*) studies were controversial about whether orexin A could activate dopaminergic neurons in the SN. Since Ox1R and Ox2R are expressed by non-dopaminergic neurons around the SN, recordings in the absence of synaptic blockers, could detect a mixed direct and indirect orexin-induced response in dopaminergic neurons (*Nair-Roberts et al., 2008*). This may partially explain their controversial findings.

We found that Ox1R deficiency in dopaminergic neurons caused an increase in novelty-induced locomotion and exploration, which could indicate elevated arousal levels and hyperactivity. Interestingly, children with attention deficit and hyperactivity disorder (ADHD) have lower orexin A but unaltered orexin B levels in plasma compared to the healthy controls (*Baykal et al., 2019*). ADHD patients exhibit impaired dopaminergic functions, and medicine improving dopaminergic signaling is used to treat ADHD in clinics (*Feldman and Reiff, 2014*; *Volkow and Swanson, 2013*). Ox1R signaling in dopaminergic neurons is important to limit novelty-induced arousal and hyperactivity, which are related to task performance and learning capability. Novelty-induced behavioral response should be at proper levels under normal physiological conditions. The orexin-dopamine interaction blunting novelty-induced locomotion could be important to keep attention on the main task without

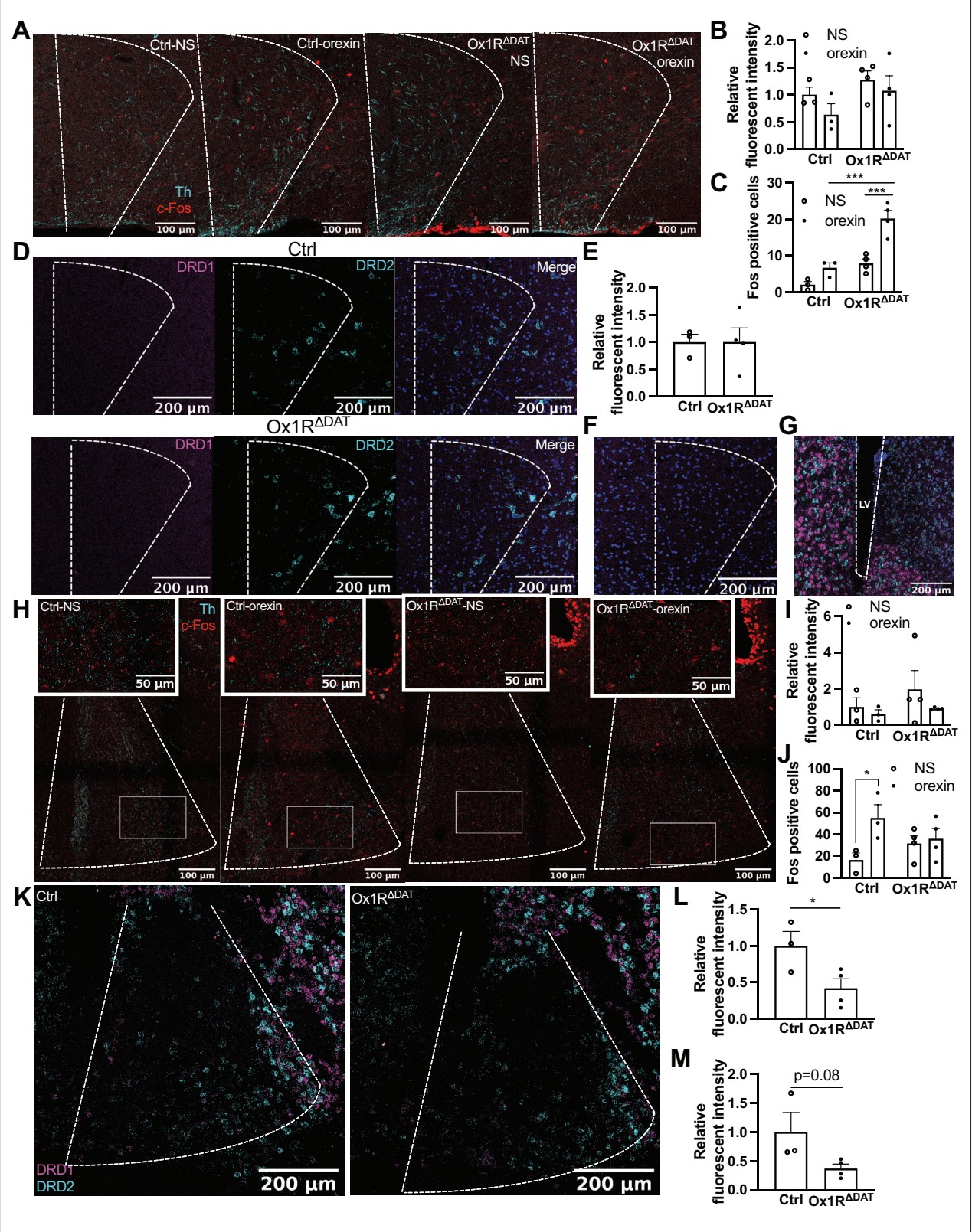

**Figure 4.** c-Fos and dopamine receptors in the lateral paragigantocellular nucleus (LPGi) and the dorsal bed nucleus of the stria terminalis (BNST). (**A**) Representative images of c-Fos and tyrosine hydroxylase (Th) staining in LPGi of control and Ox1R$^{\Delta DAT}$ mice injected (ICV) with saline (NS) and orexin A. Quantification of (**B**) Th fluorescence and (**C**) c-Fos positive neurons in LPGi. (**D**) Representative images of D1 and D2 subtypes of dopamine receptor (DRD1 and DRD2) in LPGi of control and Ox1R$^{\Delta DAT}$ mice. (**E**) Quantification of DRD2 fluorescence in LPGi. (**F**) Representative images of a negative control

*Figure 4 continued on next page*

*Figure 4 continued*

staining of DRD1 and DRD2 in LPGi of control mice, and (**G**) a positive control staining around the lateral ventricle (LV). (**H**) Representative images of c-Fos and Th staining in the dorsal BNST of control and Ox1R^ΔDAT mice injected (ICV) with saline or orexin A. Quantification of (**I**) Th fluorescence and (**J**) c-Fos positive neurons in dorsal BNST. (**K**) Representative images of DRD1 and DRD2 in the dorsal BNST of control and Ox1R^ΔDAT mice. In the interest of clarity, the representative images for single channel signal are shown in the **Figure 4—figure supplement 3**. Quantification of (**L**) DRD1 and (**M**) DRD2 fluorescence in dorsal BNST. Scale bar: 200 μm (**D, F, G, K**), 100 μm (**A, H**), or 50 μm (insertions in H). Control, n=3; Ox1R^ΔDAT, n=4. Cyan, Th; red, c-Fos (**A, H**). Magenta, DRD1; cyan, DRD2; blue, dapi (**D, F, G, K**). Data are represented as means ± SEM. *p<0.05; ***p<0.001; as determined by unpaired two-tailed Student's t-test (**E, L, M**), or two-way ANOVA followed by Sidak's post hoc test (**B, C, I, J**). (**B**) Treatment: $F_{(1, 10)}$=1.76, p=0.21; genotype: $F_{(1, 10)}$=2.83, p=0.12; treatment × genotype: $F_{(1,10)}$ = 0.15, p=0.71; (**C**) treatment: $F_{(1, 10)}$=27.94, p=0.0004; genotype: $F_{(1, 10)}$=36.42, p=0.0001; treatment × genotype: $F_{(1,10)}$ = 5.72, p=0.04; (**I**) treatment: $F_{(1, 10)}$=1.24, p=0.29; genotype: $F_{(1, 10)}$=0.95, p=0.35; treatment × genotype: $F_{(1,10)}$=0.26, p=0.62; (**J**) treatment: $F_{(1, 10)}$=5.76, p=0.04; genotype: $F_{(1, 10)}$=0.05, p=0.83; treatment × genotype: $F_{(1,10)}$=3.65, p=0.09.

The online version of this article includes the following figure supplement(s) for figure 4:

**Figure supplement 1.** A schematic illustration of the dorsal bed nucleus of the stria terminalis (dBNST) and lateral paragigantocellular nucleus (LPGi) based on the reference atlas.

**Figure supplement 2.** Dopaminergic fibers innervating the dorsal bed nucleus of the stria terminalis (BNST) and lateral paragigantocellular nucleus (LPGi).

**Figure supplement 3.** Representative images showing dopamine receptors in the dorsal bed nucleus of the stria terminalis (BNST).

being distracted too much by other random environmental stimuli. When this balance is disrupted, the behavioral deficits may happen, such as ADHD.

We did not observe a significant impact of Ox1R signaling in dopaminergic neurons on reward processing. Previous studies reported that Ox1R signaling in VTA promoted reward processing with pharmacological methods (*Tsujino and Sakurai, 2013*). The difference could have resulted from the involvement of non-dopaminergic neurons in VTA or different behavioral models. It would be interesting to identify the effect under other conditions, such as obesity and operant reward learning, in the future (*Berke, 2018*; *Dabney et al., 2020*; *Kim et al., 2020*; *Wang et al., 2001*).

The PET imaging results suggest that Ox1R signaling in dopaminergic neurons affects separate and distinct neuronal circuits, depending on the stimulation mode. Orexin and dopaminergic systems are known to increase arousal in mammals' sleep-wake transitions, locomotor activity, and motivation (*Berke, 2018*; *Berridge et al., 2010*; *Furlong et al., 2009*; *Mahler et al., 2014*). The orexin system responds to various environmental stimuli, such as stress and novelty (*Berridge et al., 2010*; *González et al., 2016*). Projection-defined dopaminergic populations in the VTA exhibited different responses to orexin A (*Baimel et al., 2017*). Our PET imaging experiments, without orexin application, indicate that Ox1R signaling in dopaminergic neurons modulates brain regions (MPA, Pir, endopiriform claustrum, LSS, SubCV, spinal trigeminal nucleus, sp5, IRt, and Gi) important for olfaction, eating, arousal and sleeping (*Supplementary file 1*). When central orexin levels were increased by orexin A injection, Ox1R signaling in dopaminergic neurons regulates HDB, MCPO, Pir, DEn, S1, V2L, V1, BNST, LPGi, and M2, which are important for sensation, emotion, exploration, and action selection (*Supplementary file 1*). In future studies, it would be interesting to investigate the direct and indirect downstream pathways of Ox1R-expressing dopaminergic neurons. Interestingly, in the fruitfly *Drosophila melanogaster*, it has been shown that the dopaminergic system oppositely regulates sleep-wake arousal and environmentally stimulated arousal, and the authors discussed the possibility of identify similar patterns in mammals (*Lebestky et al., 2009*). It was further suggested that the dopaminergic system differentially regulates arousal levels under different stimulating conditions (*Lebestky et al., 2009*; *Lima and Miesenböck, 2005*). Emotion perception affects the decision of how to respond to the novelty. We think that novelty activates the orexin system, and Ox1R signaling in dopaminergic neurons promotes emotion perception and inhibits exploration.

DRD1 and DRD2 in the BNST regulate anxiety, fear, and addiction (*De Bundel et al., 2016*; *Eiler et al., 2003*; *Giardino et al., 2018*; *Krawczyk et al., 2013*). LGPi regulates exploratory behaviors and locomotor speed (*Takatoh et al., 2013*; *Arber and Costa, 2022*; *Capelli et al., 2017*). We found a potential direct impact of dopaminergic neurons on LPGi and BNST. The inhibitory D2 receptor was mainly expressed in LPGi, and the expression levels of the excitatory D1 receptor were decreased in BNST after Ox1R deletion in dopaminergic neurons. Our findings suggested that Ox1R deletion in dopaminergic neurons results in the disinhibition of neural activity in LPGi via dopaminergic pathways and the decrease of dopamine-mediated neural activity in BNST. The expression levels of Th were

not altered in dBNST or LPGi by Ox1R deletion in dopaminergic neurons. It remains unclear whether dopamine release is affected in these regions. It is possible that either the dopaminergic regulation of neuronal activity or the changes in dopamine release could lead to the decreased expression of dopamine receptors in dBNST. It would be very interesting to further investigate these pathways in the future. There are few publications on the functional relevance of dopaminergic projections in the LPGi. Future studies would be very interesting in terms of functionally analyzing DRD2 signaling in the LPGi and how orexin interacts with DRD2 signaling.

## Materials and methods

### Animal care

All animal procedures were conducted in compliance with protocols approved by the local government authorities (Bezirksregierung Cologne, Germany; reference number: 84–02.05.40.14.134) and were in accordance with National Institutes of Health guidelines. Mice were housed in groups of 3–5 at 22°C–24°C using a 12 hr light/12 hr dark cycle. Animals had ad libitum access to water and food at all times. Animals were fed a NCD (V1554, ssniff$^R$), which contains 67% calories from carbohydrates, 23% calories from protein, and 10% calories from fat.

### Generation of Ox1R$^{\Delta DAT}$ mice

Mice carrying a loxP-flanked Ox1R-allele (Ox1R$^{fl/fl}$) were generated in our facility (*Xiao et al., 2021*). Ox1R$^{fl/fl}$ mice were crossed with Dat-Cre $^{tg/wt}$ mice (*Ekstrand et al., 2007*) to obtain Ox1R$^{fl/fl}$, Dat-Cre$^{tg/wt}$ mice, i.e., dopaminergic-specific Ox1R knockout mice on a C57BL/6 N background (Ox1R$^{\Delta DAT}$). Mice from the same breeding without the Cre transgene were used as controls (Ox1R$^{fl/fl}$, Dat-Cre$^{wt/wt}$).

### Glucose and insulin tolerance tests

Glucose tolerance tests were performed on 12-wk-old animals that had been fasted for 6 hr from 9 am. Insulin tolerance tests were performed on 11-wk-old random-fed mice around 9 am. Animals received an intraperitoneal injection of 20% glucose (10 ml/kg body weight; KabiPac) or insulin (0.75 U/kg body weight; Sanofi-Aventis) into the peritoneal cavity, respectively. Blood glucose values were determined in whole venous blood using an automatic glucose monitor (Contour$^R$, Bayer). Glucose levels were determined in blood collected from the tail tip immediately before and 15, 30, and 60 min after the injection, with an additional value determined after 120 min for the glucose tolerance tests.

### Analysis of body composition

Nuclear magnetic resonance (NMR) was employed to determine the whole body composition of live animals on NCD at the age of 20 wk using the NMR Analyzer minispec mq7.5 (Bruker Optik, Ettlingen, Germany).

### Histology

Brains from male mice were removed, post-fixed in 4% paraformaldehyde (PFA) for the indicated time, and dehydrated in 20% sucrose in 0.1 M phosphate-buffered saline (PBS) overnight. Brains were stored at –80 °C until cutting.

#### Immunostaining

Brains were post-fixed for 6 hr at 4 °C and cut (30 μm) with Leica CM3050 S Research Cryostat. Sections were incubated in 0.3% glycine for 10 min after washing in 0.1 M PBS for 2×10 min. After washing in PBS for another 10 min, sections were incubated in 0.03% SDS (in PBS) for 10 min before they got blocked with 3% donkey serum (in PBS, 0.25% Triton X-100) for 1 hr at RT. Afterward, they were incubated with primary antibodies, including goat anti-c-Fos (sc-526, Santa Cruz Biotechnology), chicken anti-GFP (ab13970, Abcam), and/or rabbit anti-Th (ab112, Abcam), for overnight (GFP) or 48 hr (c-Fos and Th) at 4 °C, washed in PBS for 3×10 min, and incubated with secondary antibodies, including Alexa Fluro 594 donkey anti-goat, FITC donkey anti-chicken, Alexa Fluro 488 donkey anti-rabbit (a11058, sa 1–7200 and a21206, Invitrogen), for 1 hr at RT. After washing in PBS for 3×10 min, slices were mounted and covered with VECTASHIELD Antifade Mounting Medium with DAPI (Vector Laboratories). Slices were stained together at one time for each experiment to have identical conditions for

comparable signals. Slides were stored at 4 °C until imaging. Images were obtained with a confocal laser scanning microscope Leica TCS SP8.

## RNAscope fluorescent in situ hybridization

Brains from 12-wk-old male mice were post-fixed for 20–22 hr at RT and cut (20 μm). Slides were stored at –80 °C until staining. All reagents were purchased from Advanced Cell Diagnostics, and the staining was performed with the RNAscope Multiplex Fluorescent v2 kit (ACD, Advanced Cell Diagnostics) according to the user manual. Probes for Ox1R (*Hcrtr1*, 471561-C3, ACD) and Ox2R (*Hcrtr2*, 471551, ACD), DRD1 (*Drd1a*, 406491-C2, ACD), DRD2 (*Drd2*, 406501), and Th (*Th*, 317621-C4) were purchased from ACD. In brief, slides were briefly washed in diethyl-pyrocarbonate (DEPC)-treated Millipore water, air dried, and then dried at 60 °C overnight. On the second day, slides were treated with hydrogen peroxide ($H_2O_2$) for 10 min at RT, washed in water, and boiled in Target Retrieval solution (around 99.4 °C) for 8–10 min. After a brief washing in water and dehydration in absolute ethanol, slides were incubated with protease IV for 30 min at RT. Slides were washed again in water and hybridized with the mixture of probes in different channels for 2 hr in a humidified chamber at 40 °C. Afterward, the hybridization was amplified with AMP 1 for 30 min, AMP 2 for 30 min, and AMP 3 for 15 min. The signal was then developed for each channel. For example, for channel 1, slides were incubated with HRP-C1 for 15 min, the fluorophore for 30 min, and the HRP blocker for 15 min. All amplification and development were performed at 40 °C, and 2×2 min of washing in ACD wash buffer was performed after each step. Finally, the slides were counterstained with DAPI for 1 min, mounted with Prolong Gold Antifade reagent with DAPI (Invitrogen), and covered with coverslips.

Slides were dried and stored at 4 °C. Imaging was performed with a confocal laser scanning microscope Leica TCS SP8.

## Quantification

The images were manually analyzed with Image J/FIJI (version 1.50d). Dopaminergic cells were manually identified and outlined based on Th and DAPI signals. The integrated intensity of Ox1R and Ox2R signal in the delineated cells was automatically calculated by Image J. The integrated intensity in the negative control area was used as a threshold, so that the cells with integrated intensity higher than the threshold were identified as Ox1R- or Ox2R-positive dopaminergic cells. c-Fos positive cell numbers were manually counted. Th, DRD1, and DRD2 fluorescence in the respective areas was taken as the integrated intensity automatically calculated by Image J.

## Ca²⁺ imaging and electrophysiology

### Animals and brain slice preparation

Perforated patch clamp recordings and $Ca^{2+}$ imaging of dopaminergic SN neurons in acute brain slices were carried out essentially as described in *Hess et al., 2023* and *Xiao et al., 2021*. $Ca^{2+}$ imaging experiments were performed on brain slices from 18 to 19 wk-old male control and Ox1R$^{\Delta DAT}$ mice. Animals were kept under standard laboratory conditions, with tap water and chow available ad libitum, on a 12 hr light/dark cycle. The animals were lightly anesthetized with isoflurane (B506; AbbVie Deutschland GmbH and Co KG) and decapitated. Coronal slices (280 μm) containing the SN were cut with a vibration microtome (VT1200 S; Leica) under cold (4 °C), carbogenated (95% $O_2$ and 5% $CO_2$), glycerol-based modified artificial cerebrospinal fluid (GaCSF). GaCSF contained (in mM): 244 Glycerol, 2.5 KCl, 2 $MgCl_2$, 2 $CaCl_2$, 1.2 $NaH_2PO_4$, 10 HEPES, 21 $NaHCO_3$, and 5 Glucose adjusted to pH 7.2 with NaOH. Brain slices were transferred into carbogenated artificial cerebrospinal fluid (aCSF). First, they were kept for 20 min in a 35 °C 'recovery bath' and then stored at room temperature (24 °C) for at least 30 min before recording. During the recordings, the brain slices were continuously superfused with carbogenated aCSF at a flow rate of ~2.5 ml·min⁻¹. aCSF contained (in mM): 125 NaCl, 2.5 KCl, 2 $MgCl_2$, 2 $CaCl_2$, 1.2 $NaH_2PO_4$, 21 $NaHCO_3$, 10 HEPES, and 5 Glucose adjusted to pH 7.2 with NaOH. To block GABAergic and glutamatergic synaptic input, the aCSF contained $10^{-4}$ M PTX (picrotoxin, P1675; Sigma-Aldrich), $5×10^{-6}$ M CGP (CGP-54626 hydrochloride, BN0597, Biotrend), $5×10^{-5}$ M DL-AP5 (DL-2-amino-5-phosphonopentanoic acid, BN0086, Biotrend), and $10^{-5}$ M CNQX (6-cyano-7-nitroquinoxaline-2,3-dione, C127; Sigma-Aldrich). Orexin A (ab120212, Abcam) was bath-applied via the superfusion system at concentrations of 100 nM and 300 nM for 8 min for each concentration.

## Perforated patch clamp recordings

Perforated patch clamp recordings were performed in the current clamp mode. The experiments were carried out using protocols modified from previous studies, as summarized in *Hess et al., 2023*. SN dopaminergic neurons were identified according to their sag component/slow $I_h$-current (hyperpolarization-activated cyclic nucleotide-gated cation current), broad action potentials (*Lacey et al., 1989*; *Neuhoff et al., 2002*; *Richards et al., 1997*) and post hoc by biocytin-streptavidin labeling combined with TH-immunohistochemistry (*Hess et al., 2013*). The electrode solution contained (in mM): 140 K-gluconate, 10 KCl, 10 HEPES, 0.1 EGTA, 2 MgCl₂, and 1% biocytin (B4261, Sigma) adjusted to pH 7.2 with KOH. The calculated liquid junction potential between intracellular and extracellular solution was compensated. The patch electrode was tip-filled with electrode solution and backfilled with electrode solution, which contained the ionophore amphotericin B (A4888; Sigma) to achieve perforated patch recordings, 0.02% tetramethylrhodamine-dextran (3000 MW, D3308, Invitrogen) to monitor the stability of the perforated membrane, and 1% biocytin (B4261; Sigma-Aldrich) to label the recorded neuron. Amphotericin B was dissolved in dimethyl sulfoxide (DMSO; D8418, Sigma) to a concentration of 200 µg*µl⁻¹ and added to the electrode solution. The ionophore was added to the modified electrode solution shortly before use. The final concentration of amphotericin B was ~120–160 µg*ml⁻¹.

The orexin A (ab120212, Abcam) effect was analyzed by comparing the action potential frequencies measured during 4- min intervals recorded before and at the end of the peptide applications. To analyze the orexin A responsiveness, the neuron's firing rate averaged from 10 s intervals was taken as one data point. To determine the mean firing rate and standard deviation, 24 data points were averaged. On the single-cell level, a neuron was considered orexin-responsive if the change in firing induced by orexin A was three times larger than the standard deviation (3 σ criterion ≙ Z-score of 3). The significance of the mean response (Δfrequency) was tested using a one-sample Wilcoxon signed rank test. Data analysis was performed using Spike2 (version 7; Cambridge Electronic Design Ltd.), Igor Pro 6 (Wavemetrics), and Prism 8 (GraphPad Software Inc).

## Ca²⁺ imaging

All imaging experiments were performed on acute brain slices using the genetically encoded calcium indicator GCaMP6s. At least 4 wk were allowed for GCaMP6s virus expression. The imaging setup consisted of a Zeiss AxioCam/MRm CCD camera with a 1388x1040 chip and a Polychromator V (Till Photonics) that was coupled via an optical fiber into the Zeiss AxioExaminer upright microscope (Objective W 'Plan-Apochromat' 20 x/1.0 DIC D=0.17 M27 75 mm). The camera and polychromator were controlled by the software Zen Pro, including the module 'Physiology' (2012 blue edition, Zeiss). The SN neurons were identified according to their anatomical location and expression of the GCaMP6s. Calcium signals in GCaMP6s expressing cells were monitored by images acquired at 470 nm excitation wavelengths with 20 ms exposure time at ~0.2 Hz. The emitted fluorescence was detected through a 500–550 nm bandpass filter (BP525/50), and data were acquired using 4×4 on-chip binning. Images were recorded in arbitrary units (AU) and analyzed as 16-bit grayscale images. Orexin A (ab120212, Abcam) in the concentrations of 100 nM and 300 nM was applied for 8 min each. To analyze the orexin A effect, we compared the fluorescence measured in the cell bodies during 4- min intervals that were recorded immediately before and at the end of the peptide applications. This protocol was followed by applying high K⁺ (40 mM) concentration saline (40 mM KCl; osmolarity adjusted by reducing the NaCl concentration) to identify cell bodies of neurons with functional GCaMP6s expression. Regions of interest (ROI) were defined after the experiment based on the high K⁺ saline responses. The mean AU values of the ROIs were calculated in Image J. Time series analysis was performed with Igor Pro 6. To correct for bleaching artifacts, the baseline fluorescence (without orexin application) was fit. The extended fit was subtracted from the raw data. For each neuron, we tested whether it responded to orexin A. A neuron was considered orexin-responsive if the change in GCaMPs fluorescence induced by orexin A was three times larger than the standard deviation (3 σ criterion ≙ Z-score of 3) of the baseline fluorescence. To determine the 'population response' of all analyzed neurons, we measured the orexin-induced GCaMP6s fluorescence of each neuron expressed as a percentage of the GCaMP6s fluorescence induced by the high (40 mM) K⁺ saline application (*Figure 1—figure supplement 3*). This is a solid reference point since it reflects the GCaMP6s fluorescence at maximal voltage-activated Ca²⁺ influx. The 'population response' of all analyzed neurons was expressed as the mean ± SEM of these

responses. The significance of this mean response was tested for each group (control and Ox1R$^{\Delta DAT}$) using a one-sample t-test. Image analysis was performed offline using Image J (version 1.53 a), Igor Pro 6 (WaveMetrics), and Prism 8 (GraphPad Software Inc).

## Food intake and spontaneous activity

Mice (20-wk-old) were analyzed for food intake, spontaneous activity, respiratory exchanging ratio, and energy expenditure in a PhenoMaster System (TSE Systems) as previously described (*Jordan et al., 2011*).

## Behavioral studies

Mice were moved into the behavioral test room, weighed, and handled 1 wk before the measurement. On the day of the tests, mice (9–15 wk old) were moved near to the equipment and handled 1 hr before the tests. There was a 1 wk break between different behavioral tests for anxiety, exploration, or locomotion when using the same mice.

### Open field test (OFT)dark/light (D/L) box, and hole-board tests

OFT and D/L box tests were performed with a Seamless Open Field Starter Package for the Mouse plus the Light/Dark Insert or the hole-board floor (Med Associates, Inc) between 10 am to 4 pm.

To start OFT, mice were put in the middle of a clear Plexiglas chamber (27.3 cm×27.3 cm×20.3 cm) sitting in a sound-attenuating cubicle, and mouse movement was automatically traced for 30 min with 48 infrared beams divided into the X, Y, and Z plane. For the D/L box test, a black Plexiglas box (half the size of the chamber) was inserted at the left side of the chamber, with an opening at the middle side. Mice were put in the middle of the light side and faced the openings. Mouse movement was traced for 5 min. For the hole-board test, a metal hole-board floor with 16 equally spaced holes for exploration was inserted in the chamber. Mice were put in the middle of the chambers and the nose pokes were monitored automatically for 10 min per session. There was 1 hr break between each session. Four sessions per day were analyzed for two consecutive days.

### Elevated 0-maze test

An elevated 0-maze test was performed with TSE elevated 0-Maze in combination with the video tracking system TSE VideoMot 3D version 7.01. There are two closed and two open runways without a center position. The outer diameter is 48 cm, the arm width is 5 cm, the wall height is 10 cm, and the base height is 55 cm. Mice were put in the middle of an open arm and faced to the closed arm to start the experiment. Mouse movement was monitored for 5 min.

### Novel object recognition test

Novel object recognition test was modified from the publication before (*Ruud et al., 2019*). The mouse was acclimated and measured for 10 min in the first session and for 5 min in the second session on the same day, in a polycarbonate box with non-transparent walls (50 cm×50 cm×30 cm). Two objects were present in the box. In the first session, two identical objects (glass beaker filled with blue marbles) were present and fixed with odor-free adhesive to the ground flooring at two sides of the box delineated diagonally. In each diagonal corner, the object is 11.5 cm and 14 cm from each wall, respectively, and the distance between the two objects is 17.5 cm. Mice always started at the same corner which was between the two fixed objects and was allowed to explore and learn the experimental area and objects. In the second session which started after a 1 hr break in the home cage, one of the two identical objects was replaced with a novel object (Lego Primo brick), and mice were allowed again to explore the environment and objects. The positions of the mouse nose, tail, and center were monitored with a video tracking system TSE VideoMot 2. The interaction with a fixed object was identified when the nose was facing the object within a distance of 2 cm.

### Conditioned place preference test for cocaine

The test was modified from previous publications (*Ruud et al., 2019*), and was performed in a polycarbonate box with non-transparent walls. The box was separated into three compartments with two independent partitions. It contains a left compartment with zebra stripes (white and black) painted

on the wall and a rough ground (16.5×13.2×21.2 cm), a small middle area (7.0×13.2×21.2 cm), and a right compartment with a white wall and smooth ground. Mouse movement was monitored with TSE VideoMot 2 for 30 min per session, one session per day. Mice started from the middle area and had access to both left and right compartments through an opening (4.0 cm×4.2 cm) in the partition during the habituation and pre-test days (d1-3) and on the test day (d10), and were put into and kept in either the left or right compartment without an opening in the partition during the conditioning phase (d4-9). The preference of the two compartments by nature was analyzed during the pre-test days. At the conditioning phase, on alternative days, mice received an i.p. injection of cocaine (5 mg/kg BW) or saline immediately before entering the respective compartments. Cocaine and saline were always paired to the non-preferred and preferred compartments, which were determined by pre-tests, respectively. There was no injection on the habituation, pre-test, or test days. The percentage of time spent in the reward-paired compartment was calculated: 100 x time spent in the compartment/(total time - time spent in the middle area). The CPP score was then analyzed using the calculated percentage of time: 100 x (time on the test day – time on pre-test days)/time on pre-test days. The pre-test and test days were before and after the conditioning, respectively. Thus, the CPP score above zero indicates that the CPP preference is developed.

## Sucrose/sucralose preference test

Two-bottle tests were performed with two 15 ml falcon tubes connected to sippers. Mice were single-caged 1 wk before the beginning of the experiment. Two falcon tubes filled with water (side-by-side) replaced the normal water bottle. Water intake was monitored for a week to make sure mice adjusted to drinking from the sippers. When the test started, two new sipper tubes of water were used for 2 d of baseline measurement and were then replaced with one containing water and one containing sucrose/sucralose solution for 2 d of preference measurement. Animals received a 0.5%, 1%, and 2% (w/v) sucrose solution and 1 mM sucralose solution in sequential. Between the preference tests of different solutions, 2 d of baseline measurement were performed with two new sipper tubes of water as a washout. To avoid side bias, the positions of two sipper tubes were switched every day, with the sucrose/sucralose tube always starting on the non-preferred position, which was determined by water intake on the day before the preference measurement. Sipper tubes were weighed every day.

## ICV cannula implantation and injection

12-wk-old mice received 1 mg/ml of tramadol in drinking water 2 d before the surgeries. On the day of surgery, mice were anesthetized with isoflurane, and a bolus of buprenorphine (0.1 mg/kg BW) was given (i. p.) to reduce pain. Brain regions were identified according to the atlas (*Franklin and Paxinos, 2007*). After alignment of the brain in the stereotaxic surgery platform, one small hole was drilled in the skull and the ICV cannula (26 gauge, Plastics One) was immediately implanted (AP: –0.2 mm, ML: +1 mm, DV: –2.4 mm) and fixed to the skull with dental acrylic (Super-Bond C&B). The dummy cannula caps were inserted to close and protect the cannula. Mice received a bolus of meloxicam (5 mg/kg BW, s.c.), and tramadol in drinking water for 3 d after surgeries to relieve pain. BW was checked twice a day. Experiments started at least 1 wk later. For injection, the catheter was attached to an injector (Plastics One), which was inserted through the cannula. Orexin A (Phoenix Pharmaceuticals) was injected slowly at the dose of 1 nmol dissolved in 2 ul of saline. Starting from 5 d before the injection, mice and the dummy cannula caps were handled for acclimation daily. The locomotion activity was measured in a PhenoMaster System (TSE Systems), and mice were acclimated to the system at least 5 d before the test.

## PET imaging

PET imaging was performed using an Inveon preclinical PET/CT system (Siemens). Male mice were anesthetized with 2% isoflurane in 65%/35% nitrous oxide/oxygen gas, injected with saline or orexin A (Phoenix Pharmaceuticals), and positioned on a dedicated mouse carrier (MEDRES, Germany) carrying two mice. Body temperature was maintained at 37.0 ± 0.5°C by a thermostatically controlled water heating system. For injection of the radiotracer, a catheter consisting of a 30 G cannula connected to a polythene tubing (ID = 0.28 mm) was inserted into the tail vein and fixated by a drop of glue. After starting the PET scan, 7–8 MBq of [18 F]FDG in 50–100 µl saline were injected per mouse. Emission data were acquired for 45 min. Thereafter, animals were automatically moved into the CT gantry, and

a CT scan was performed (180 projections/360°, 200 ms, 80 kV, 500 µA). The CT data were used for attenuation correction of the PET data, and the CT image of the scull was used for image co-registration. Plasma glucose levels were determined from a tail vein blood sample using a standard glucometer (Bayer) after removing the tail vein catheters. PET data were histogrammed in time frames of 12×30 s, 3×60 s, 3×120 s, 7×240 s, Fourier rebinned, and images were reconstructed using the MAP-SP algorithm provided by the manufacturer. For co-registration, the imaging analysis software Vinci was used (*Cízek et al., 2004*). Images were co-registered to a 3D mouse brain atlas constructed from the 2D mouse brain atlas published by Paxinos (*Paxinos et al., 2013*).

## Kinetic modeling

An image-derived input function was extracted from the PET data of the aorta, which could be identified in the image of the first time frame of each animal. Input function data were corrected for partial volume effect by assuming a standardized volume fraction of 0.6 (*Green et al., 1998*). Parametric images of the [18 F]FDG kinetic constants $K_1$, $k_2$, $k_3$, and $k_4$ were determined by a voxel-by-voxel (voxel size = 0.4 mm×0.4 mm×0.8 mm) fitting of data to a two- tissue-compartment kinetic model. $K_1$ is the constant for transport from blood to tissue, $k_2$ for transport from tissue to blood, $k_3$ the constant for phosphorylation of [18 F]FDG to [18 F]FDG-6-phosphate, and $k_4$ the constant of dephosphorylation. The ratio of tissue and plasma glucose concentrations ($C_E/C_P$) is a measure for glucose transport and is given by $C_E/C_P = K_1/(k_2 + k_3/0.26)$ (*Backes et al., 2011*; *Jais et al., 2016*). Since neuronal activation is accompanied by increased glucose transport and this parameter is less sensitive to changes in plasma glucose level, we use alterations of glucose transport ($C_E/C_P$) as a surrogate for alterations in neuronal activation.

## Statistics

Statistical testing was performed by application of a voxel-wise t-test between groups. 3D maps of p-values allow for the identification of regions with significant differences in the parameters. From these regions, we defined volumes of interest (VOIs) and performed additional statistical testing for these VOIs. For presentation only, 3D maps of p-values were re-calculated on a 0.1 mm×0.1 mm×0.1 mm grid from the original dataset using trilinear interpolation.

## Virus injection

Male mice received 1 mg/ml of tramadol in drinking water 2 d before the surgeries. On the day of surgery, mice were anesthetized with isoflurane, and a bolus of buprenorphine (0.1 mg/kg BW) was given (i. p.) to reduce pain. Brain regions were identified according to the atlas (*Franklin and Paxinos, 2007*). After alignment of the brain in the stereotaxic surgery platform, one or four small holes were drilled in the skull at specific coordinates. The virus, pAAV-Ef1a-DIO EYFP (~1×10^13 GC/ml, 300 nl, Addgene) was bilaterally injected into VTA and SN of 10-wk-old mice, with micropipettes pulled in-house with a heating system. The coordinates from Bregma were: anterior-posterior, AP: –3.4 mm; medial-lateral, ML: ±0.5/1.25 mm; dorsal-ventral, DV: –4.2 mm. The GCaMP6s virus (AAV1. Syn.Flex.GCaMP6s.WPRE.SV40, ~4×10^12 GC/ml, 300 nl, Penn Vector Core) was injected to left SN (AP: –3.4 mm, ML: 1.1 mm, DV: –4.2 mm) of 12-wk-old mice, to specifically express GCaMP6s in dopaminergic neurons. After virus injection, mice received a bolus of meloxicam (5 mg/kg BW, s.c.), and tramadol in drinking water for 3 d after surgeries to relieve pain. BW was checked twice a day. Experiments started at least 4 wk later than virus injection to allow virus expression.

## Statistical methods

All numerical values are expressed as the mean ± SEM. Statistical analyses were conducted using GraphPad PRISM (version 8) unless stated otherwise. Datasets with only two independent groups were analyzed for statistical significance using an unpaired two-tailed Student's t-test. Datasets subjected to two independent factors were analyzed using two-way ANOVA followed by Sidak's post hoc test. All p-values <0.05 were considered significant (*p<0.05, **p<0.01, and ***p<0.001, ****p<0.0001).

## Acknowledgements

We acknowledge Hella Brönneke for outstanding support, Jens Alber, Pia Scholl, Christiane Schäfer, Nadine Spenrath, Ina Stünkel, Kerstin Marohl, Antonia Lorenz, and Helmut Wratil for outstanding technical assistance. We received funding by the DFG within the framework of the TRR 134 (ACH, PK), DFG-401832153 (PK), and within the Excellence Initiative by German Federal and State Governments (CECAD). This work was funded (in part) by the Helmholtz Alliance ICEMED (Imaging and Curing Environmental Metabolic Diseases) through the Initiative and Networking Fund of the Helmholtz Association. XX received funding from CECAD Family Support. GY gratefully acknowledges financial doctoral support from the DFG-233886668/GRK1960.

## Additional information

### Funding

| Funder | Grant reference number | Author |
| --- | --- | --- |
| Deutsche Forschungsgemeinschaft | TRR 134 | Peter Kloppenburg Anne Christine Hausen |
| Deutsche Forschungsgemeinschaft | 401832153 | Peter Kloppenburg |
| Helmholtz Association | | Anne Christine Hausen |
| Deutsche Forschungsgemeinschaft | 233886668/GRK1960 | Gagik Yeghiazaryan |

The funders had no role in study design, data collection and interpretation, or the decision to submit the work for publication. Open access funding provided by Max Planck Society.

### Author contributions

Xing Xiao, X.X. conceived the project, designed the experiments, performed all the experiments apart from calcium imaging and electrophysiological recording experiments and PET imaging experiments, analysed data and wrote the manuscript with input from the other authors; Gagik Yeghiazaryan, G.Y. performed and analysed calcium imaging and electrophysiological recording experiments; Fynn Eggersmann, F.E. performed and analysed electrophysiological recording experiments; Anna Lena Cremer, A.L.C. performed PET imaging experiments; Heiko Backes, H.B. supervised, performed and analysed the PET imaging experiments; Peter Kloppenburg, P.K. supervised the calcium imaging and electrophysiological recording experiments as well as manuscript writing; Anne Christine Hausen, A.C.H. conceived the project, designed the experiments, generated the Ox1R flox mouse line, and supervised the projects and manuscrpt writing

### Author ORCIDs

Xing Xiao https://orcid.org/0000-0003-3590-7430
Peter Kloppenburg https://orcid.org/0000-0002-4554-404X

### Ethics

All animal procedures were conducted in compliance with protocols approved by the local government authorities (Bezirksregierung Cologne, Germany; Ref 84-02.05.40.14.134) and were in accordance with National Institutes of Health guidelines.

Reviewer #1 (Public review): https://doi.org/10.7554/eLife.91716.4.sa1
Reviewer #2 (Public review): https://doi.org/10.7554/eLife.91716.4.sa2
Author response https://doi.org/10.7554/eLife.91716.4.sa3

## Additional files

### Supplementary files

Supplementary file 1. Functions of brain regions.

MDAR checklist

Source data 1. Numeric data and full statistics for *Figure 1* and its related figure supplements.

Source data 2. Numeric data and full statistics for *Figure 2* and its related figure supplements.

Source data 3. Numeric data and full statistics for *Figure 3* and its related figure supplements.

Source data 4. Numeric data and full statistics for *Figure 4* and its related figure supplements.

### Data availability

All data generated or analysed during this study are included in the manuscript and supporting files; source data files have been provided.

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
