## [Editor Report · eLife Assessment]

This manuscript describes **valuable** findings regarding the expression pattern of orexin receptors in the midbrain and how manipulating this system influences several behaviors, such as context-induced locomotor activity and exploration. The overall strength of evidence - which includes anatomical, viral manipulation studies, and brain imaging - is **solid** and broadly substantiates claims in the paper. However, there are several areas in which the conclusions are only partially supported by the combination of methods used. These results have implications for understanding the neural underpinnings of reward and will be of interest to neuroscientists and cognitive scientists with an interest in the neurobiology of reward.

---

## [Referee Report · Reviewer #1 (Public review)]

In this manuscript, the role of orexin receptors in dopamine transmission is studied. It extends previous findings suggesting an interplay between these two systems in regulating behaviour by first characterizing the expression of orexin receptors in the midbrain and then disrupting orexin transmission in dopaminergic neurons by deleting its predominant receptor, OX1R (Ox1R fl/fl, Dat-Cre tg/wt mice). Electrophysiological and calcium imaging data suggest that orexin A acutely and directly stimulates SN and VTA dopaminergic neurons but does not seem to induce c-Fos expression. Behavioral effects of depleting OX1R from dopaminergic neurons include enhanced novelty-induced locomotion and exploration, relative to littermate controls (Ox1R fl/fl, Dat-Cre wt/wt). However, no difference between groups is observed in tests that measure reward processing, anxiety, and energy homeostasis. To test whether the depletion of OX1R alters overall orexin-triggered activation across the brain, PET imaging is used in OX1R∆DAT knockout and control mice. This analysis reveals that several regions show higher neuronal activation after orexin injection in OX1R∆DAT mice, but the authors focus their follow-up study on the dorsal bed nucleus of the stria terminalis (BNST) and lateral paragigantocellular nucleus (LPGi). Dopaminergic inputs and expression of dopamine receptors type-1 and -2 (DRD1 & DRD2) are assessed and compared to control demonstrating a moderate decrease in DRD1 and DRD2 expression in the BNST of OX1R∆DAT mice and unaltered expression of DRD2, with absence of DRD1 expression in LPGi of both groups. Overall, this study is valuable for the information it provides on orexin receptor expression and function in behaviour, as well as for the new tools it generated for the specific study of this receptor in dopaminergic circuits.

Strengths:

The use of a transgenic line that lacks OX1R in dopamine-transporter expressing neurons is a strong approach to dissect the direct role of orexin in modulating dopamine signaling in the brain. The battery of behavioral assays used to study this line provides valuable information for researchers interested in the interplay between dopamine and orexin systems and their role in animal physiology.

Weaknesses:

This study falls short in providing evidence for an anatomical substrate and mechanism underlying the altered behavior observed in mice lacking orexin receptor subtype 1 in dopaminergic neurons. How orexin transmission in dopaminergic neurons regulates the expression of postsynaptic dopamine receptors (as observed in the BNST of OX1R∆DAT mice) is an intriguing question not addressed in this study. An important aspect not investigated in this study is whether the disruption of orexin activity affects dopamine release in target areas.

---

## [Referee Report · Reviewer #2 (Public review)]

Summary:

This manuscript examines expression of orexin receptors in midbrain - with a focus on dopamine neurons - and uses several fairly sophisticated manipulation techniques to explore the role of this peptide neurotransmitter in reward-related behaviors. Specifically, in situ hybridization is used to show that substantia nigra dopamine neurons predominantly express orexin receptor 1 subtype and then go on to delete this receptor in dopamine transporter-expressing neurons using a transgenic strategy. Ex vivo calcium imaging of midbrain neurons is used to show that, in the absence of this receptor, orexin is no longer able to excite dopamine neurons of the substantia nigra.

The authors proceed to use this same model to study the effect of orexin receptor 1 deletion on a series of behavioral tests, namely, novelty-induced locomotion and exploration, anxiety-related behavior, preference for sweet solutions, cocaine-induced conditioned place preference, and energy metabolism. Of these, the most consistent effects are seen in the tests of novelty-induced locomotion and exploration in which the mice with orexin 1 receptor deletion are observed to show greater levels of exploration, relative to wild-type, when placed in a novel environment, an effect that is augmented after icv administration of orexin.

In the final part of the paper, the authors use PET imaging to compare brain-wide activity patterns in the mutant mice compared to wildtype. They find differences in several areas both under control conditions (i.e., after injection of saline) as well as after injection of orexin. They focus in on changes in dorsal bed nucleus of stria terminalis (dBNST) and the lateral paragigantocellular nucleus (LPGi) and perform analysis of the dopaminergic projections to these areas. They provide anatomical evidence that these regions are innervated by dopamine fibers from midbrain, are activated by orexin in control, but not mutant mice, and that dopamine receptors are present. They also show changes in receptor expression in the transgenic mice. Thus, they argue these anatomical data support the hypothesis that behavioral effects of orexin receptor 1 deletion in dopamine neurons are due to changes in dopamine signaling in these areas.

Strengths:

Understanding how orexin interacts with the dopamine system is an important question and this paper contains several novel findings along these lines. Specifically:

(1) Distribution of orexin receptor subtypes in VTA and SN is explored thoroughly.

(2) Use of the genetic model that knocks out a specific orexin receptor subtype from dopamine-transporter-expressing neurons is a useful model and helps to narrow down the behavioral significance of this interaction.

(3) PET studies showing how central administration of orexin evokes dopamine release across the brain is intriguing, especially since two key areas are pursued - BNST and LPGi - where the dopamine projection is not as well described/understood.

Weaknesses:

The role of the orexin-dopamine interaction is not explored in enough detail. The manuscript presents several related findings, but the combination of anatomy and manipulation studies do not quite tell a cogent story. Ideally, one would like to see the authors focus on a specific behavioral parameter and show that one of their final target areas (dBNST or LPGi) was responsible or at least correlated with this behavioral readout. In addition, the authors' working model for how they think orexin-dopamine interactions contribute to behavior under normal physiological conditions is not well-described.

---

## [Author Response]

The following is the authors’ response to the previous reviews.

**Public Reviews:**

**Reviewer #1 (Public review):**
In this manuscript, the role of orexin receptors in dopamine transmission is studied. It extends previous findings suggesting an interplay of these two systems in regulating behaviour by first characterising the expression of orexin receptors in the midbrain and then disrupting orexin transmission in dopaminergic neurons by deleting its predominant receptor, OX1R (Ox1R fl/fl, DatCre tg/wt mice). Electrophysiological and calcium imaging data suggest that orexin A acutely and directly stimulates SN and VTA dopaminergic neurons, but does not seem to induce c-Fos expression. Behavioural effects of depleting OX1R from dopaminergic neurons includes enhanced noveltyinduced locomotion and exploration, relative to littermate controls (Ox1R fl/fl, Dat-Cre wt/wt). However, no difference between groups is observed in tests that measure reward processing, anxiety, and energy homeostasis. To test whether depletion of OX1R alters overall orexin-triggered activation across the brain, PET imaging is used in OX1R∆DAT knockout and control mice. This analysis reveals that several regions show a higher neuronal activation after orexin injection in OX1R∆DAT mice, but the authors focus their follow up study on the dorsal bed nucleus of the stria terminalis (BNST) and lateral paragigantocellular nucleus (LPGi). Dopaminergic inputs and expression of dopamine receptors type-1 and -2 (DRD1 & DRD2) is assessed and compared to control demonstrating moderate decrease of DRD1 and DRD2 expression in BNST of OX1R∆DAT mice and unaltered expression of DRD2, with absence of DRD1 expression in LPGi of both groups. Overall, this study is valuable for the information it provides on orexin receptor expression and function on behaviour and for the new tools it generated for the specific study of this receptor in dopaminergic circuits.Strengths:The use of a transgenic line that lacks OX1R in dopamine-transporter expressing neurons is a strong approach to dissect the direct role of orexin in modulating dopamine signalling in the brain. The battery of behavioural assays to study this line provides a valuable source of information for researchers interested in the role of orexin in animal physiology.

We thank the reviewer for summarizing the importance and significance of our study.

Weaknesses:This study falls short in providing evidence for an anatomical substrate of the altered behaviour observed in mice lacking orexin receptor subtype 1 in dopaminergic neurons. How orexin transmission in dopaminergic neurons regulates the expression of postsynaptic dopamine receptors (as observed in BNST of OX1R^∆DAT^ mice) is an intriguing question poorly discussed. Whether disruption of orexin activity alters dopamine release in target areas is an important point not addressed.

We identified dopaminergic fibers and dopamine receptors in the dBNST and LPGi, suggesting anatomical basis for dopamine neurons to regulate neural activity and receptor expression levels in these areas. PET imaging scan and c-Fos staining revealed that Ox1R signaling in dopaminergic cells regulates neuronal activity in dBNST and LPGi. The expression levels of Th were unchanged in both regions. Dopamine receptor 2 (DRD2), but not DRD1, is expressed in LPGi. The deletion of Ox1R in DAT-expressing cells did not affect DRD2 expression in LPGi. The expression levels of DRD1 and DRD2 were decreased or showed a tendency to decrease in dBNST.

We included the comments in the discussion in this revised manuscript (lines 308-312): ‘The expression levels of Th were not altered in dBNST or LPGi by Ox1R deletion in dopaminergic neurons. It remains unclear whether dopamine release is affected in these regions. It is possible that either the dopaminergic regulation of neuronal activity or the changes in dopamine release could lead to the decreased expression of dopamine receptors in dBNST.’

**Reviewer #2 (Public review):**
Summary:This manuscript examines expression of orexin receptors in midbrain - with a focus on dopamine neurons - and uses several fairly sophisticated manipulation techniques to explore the role of this peptide neurotransmitter in reward-related behaviors. Specifically, in situ hybridization is used to show that dopamine neurons predominantly express orexin receptor 1 subtype and then go on to delete this receptor in dopamine transporter-expressing using a transgenic strategy. Ex vivo calcium imaging of midbrain neurons is used to show that, in the absence of this receptor, orexin is no longer able to excite dopamine neurons of the substantia nigra.The authors proceed to use this same model to study the effect of orexin receptor 1 deletion on a series of behavioral tests, namely, novelty-induced locomotion and exploration, anxiety-related behavior, preference for sweet solutions, cocaine-induced conditioned place preference, and energy metabolism. Of these, the most consistent effects are seen in the tests of novelty-induced locomotion and exploration in which the mice with orexin 1 receptor deletion are observed to show greater levels of exploration, relative to wild-type, when placed in a novel environment, an effect that is augmented after icv administration of orexin.In the final part of the paper, the authors use PET imaging to compare brain-wide activity patterns in the mutant mice compared to wildtype. They find differences in several areas both under control conditions (i.e., after injection of saline) as well as after injection of orexin. They focus in on changes in dorsal bed nucleus of stria terminalis (dBNST) and the lateral paragigantocellular nucleus (LPGi) and perform analysis of the dopaminergic projections to these areas. They provide anatomical evidence that these regions are innervated by dopamine fibers from midbrain, are activated by orexin in control, but not mutant mice, and that dopamine receptors are present. Thus, they argue these anatomical data support the hypothesis that behavioral effects of orexin receptor 1 deletion in dopamine neurons are due to changes in dopamine signaling in these areas.Strengths:Understanding how orexin interacts with the dopamine system is an important question and this paper contains several novel findings along these lines. Specifically:(1) Distribution of orexin receptor subtypes in VTA and SN is explored thoroughly.(2) Use of the genetic model that knocks out a specific orexin receptor subtype from dopaminetransporter-expressing neurons is a useful model and helps to narrow down the behavioral significance of this interaction.(3) PET studies showing how central administration of orexin evokes dopamine release across the brain is intriguing, especially that two key areas are pursued - BNST and LPGi - where the dopamine projection is not as well described/understood.

We thank the reviewer for summarizing the importance and significance of our study.

Weaknesses:The role of the orexin-dopamine interaction is not explored in enough detail. The manuscript presents several related findings, but the combination of anatomy and manipulation studies do not quite tell a cogent story. Ideally, one would like to see the authors focus on a specific behavioral parameter and show that one of their final target areas (dBNST or LPGi) was responsible or at least correlated with this behavioral readout.

We agree that exploring the orexin-dopamine interactions in more detail and focusing on the behavioral impact of their final target areas (e.g., dBNST or LPGi), would provide valuable data. While we are very interested in pursuing these studies, the aim of the present manuscript is to provide an overview of the behavioral roles of orexin-dopamine interaction and to propose some promising downstream pathways in a relatively broad and systematic manner.

In many places in the Results, insufficient explanation and statistical reporting is provided. Throughout the Results - especially in the section on behavior although not restricted to this part - statements are made without statistical tests presented to back up the claims, e.g., "Compared to controls, Ox1R^ΔDAT^ 143 mice did not show significant changes in spontaneous locomotor activity in home cages" (L143) and "In a hole-board test, female Ox1RΔDAT mice showed increased nose pokes into the holes in early (1st and 2nd) sessions compared to control mice" (L151). In other places, ANOVAs are mentioned but full results including main effects and interactions are not described in detail, e.g., in F3-S3, only a single p-value is presented and it is difficult to know if this is the interaction term or a post hoc test (L205). These and all other statements need statistics included in the text as support. Addition of these statistical details was also requested by the editor.

We submitted all our source data as Excel spreadsheets to eLife during our first-round revision, and the full statistics, such as main effects and interactions, are presented alongside the source data in the respective spreadsheets. We thank the reviewer for pointing out our lack of clarity in the manuscript. In this revised manuscript, we included the statistical details of ANOVAs mentioned above in the figure legends. In the figure legends, we also explained that the full statistics were provided alongside the source data in the supplementary materials.

In the presentation of reward processing this is particularly important as no statistical tests are shown to demonstrate that controls show a cocaine-induced preference or a sucrose preference. Here, one option would be to perform one-sample t-tests showing that the data were different to zero (no preference). As it is, the claim that "Both of the control and Ox1RΔDAT groups showed a preference for cocaine injection" is not yet statistically supported.

We thank the reviewer for the suggestions. We have added the one-sample t-test results in this revised manuscript (Figure 2–figure supplement 4, lines 171 - 183).

**Recommendations for the authors:**

**Reviewer #2 (Recommendations for the authors):**
Can the authors comment on overlap between DAT and Ox1R in brain areas outside VTA/SN? Is there any?

We only focused on the expression patterns of orexin receptors in VTA/SN, and we did not examine other brain regions. Additionally, little is known from the literature about the expression of Ox1R in DAT-expressing cells in brain areas outside VTA/SN. Further analysis is necessary to answer this question. We have added the comment in our discussion (lines 243 - 344).

For the Ca2+ imaging experiment, it is unclear to me why the authors do not show all the neurons (almost 160 in total) and just select 5 neurons to show for each condition.

Heat maps of all recorded neurons are now shown in Figure 1—figure supplement 4.

There are other claims that still require a statistical justification to be included in addition to the passages on behavior mentioned above, e.g., "Increasing the orexin A concentration to 300 nM further increased [Ca2+]i" (L118).Authors should ensure that all such claims are either presented with a statistical test or are phrased differently, e.g. "Visual inspection of data suggested that there was a further increase...". In addition, when an ANOVA is conducted, full results including main effects and interactions should be described.

We emphasize now our statement that ALREADY 100 nM orexin A significantly increased [Ca^2+^]i levels (lines 117 - 118).

We submitted all our source data as Excel spreadsheets to eLife during our first-round revision, and the full statistics, such as main effects and interactions, are presented alongside the source data in the respective spreadsheets. For clarity, we chose to include only the key statistical information in the main text and figures. We thank the reviewer for pointing this out. In this revised manuscript, we have emphasized in each figure legend: ‘Source data and full statistics are provided in the supplementary materials’.

Typos in figure captionsF2-S1 - spontanousF3-S2 - intrest

We apologize for the typos. We have corrected them in this revised manuscript.

**Editor's note:**
Should you choose to revise your manuscript, please include full statistical reporting including exact p-values wherever possible alongside the summary statistics (test statistic and df) and 95% confidence intervals. These should be reported for all key questions and not only when the p-value is less than 0.05.

We submitted all our source data as Excel spreadsheets to eLife during our first-round revision, and the full statistics, such as test statistics, df and 95% confidence intervals, are presented alongside the source data in the respective spreadsheets. We thank the editor’s note. In this revised manuscript, we have included more statistical information in the main text and figure legends (see our response to reviewer #2). In the figure legends, we also explained that the full statistics were provided alongside the source data in the supplementary materials. In addition, we also uploaded the source data and full statistics in the bioRxiv before we upload this revised manuscript to eLife.